

# Performance, usability and comparison of two versions of a new macular vision test: the handheld Radial Shape Discrimination test

Jae Y. Ku[1], Ashli F. Milling[2], Noelia Pitrelli Vazquez[1] and Paul C. Knox[1]

[1] Eye & Vision Science, University of Liverpool, Liverpool, United Kingdom
[2] Directorate of Orthoptics and Vision Science, University of Liverpool, Liverpool, United Kingdom

## ABSTRACT

**Background.** Central vision, critical for everyday tasks such as reading and driving, is impacted by age-related changes in the eye and by diseases such as age-related macular degeneration. The detection of changes in macular function is therefore important. The Radial Shape Discrimination (RSD) test measures the threshold at which distortions in a radial frequency pattern can be detected and there is evidence that it is more sensitive to macular pathology than visual acuity (VA). It also provides a more quantitative measure of macular function than the commonly available Amsler grid. Recently, handheld versions of the test (hRSD) in which stimuli are presented on mobile devices (e.g., Apple iPod Touch, iPhone) have been developed. We investigated the characteristics of the hRSD test in healthy participants.

**Methods.** Data were collected using both three-alternative forced choice (3AFC) and 4AFC versions of the hRSD test, presented on an Apple iPod Touch. For the 3AFC version, data from a single test session were available for 186 (72 male; mean $\pm$ SD age 42 $\pm$ 17y; range 16–90y) healthy participants. Test-retest data were available for subgroups of participants (intra-session: $N = 74$; tests approximately 2 months apart: $N = 30$; tests 39 months apart: $N = 15$). The 3AFC and 4AFC versions were directly compared in 106 participants who also completed a usability questionnaire. Distance and near VA and Pelli Robson Contrast Sensitivity (CS) data were collected and undilated fundoscopy performed on the majority of participants.

**Results.** Mean ($\pm$SD) 3AFC hRSD threshold was $-0.77 \pm 0.14$ logMAR, and was statistically significantly correlated with age (Pearson $r = 0.35$; $p < 0.001$). The linear regression of hRSD threshold on age had a slope of $+0.0026$ compared to $+0.0051$ for near VA (which also correlated with age: $r = 0.51$; $p < 0.001$). There were no statistically significant differences in hRSD thresholds for any of the test-retest subgroups. We also observed no statistically significant difference between 3AFC ($-0.82 \pm 0.11$ logMAR) and 4AFC ($-0.80 \pm 0.12$ logMAR) hRSD thresholds ($t = 1.85, p = 0.067$) and participants reported excellent test usability with no strong preference expressed between the 3AFC and 4AFC versions of the test.

**Discussion.** The 3AFC hRSD thresholds we report are consistent with a number of previous studies, as is its greater stability in ageing compared to VA. We have also shown that in the absence of pathology, thresholds are stable over short and long timescales. The 4AFC thresholds we have reported provide a baseline for future investigations, and we have confirmed that 3AFC and 4AFC thresholds are similar,

Corresponding author
Paul C. Knox, pcknox@liv.ac.uk

providing a basis of comparisons between studies using the different versions. As the hRSD test is easy to use and relatively inexpensive, clinical studies are now required to establish its ability to detect and monitor macular pathologies.

## INTRODUCTION

Measurement of visual acuity (VA) by means of letter charts is a standard and ubiquitous method of measuring vision for both research and clinical purposes. However, resolution acuity is only one aspect of visual function, and it is known to be relatively insensitive to the early pathological changes in retinal function that occur in important diseases such as age-related macular degeneration (*Dimitrov et al., 2011*; *Klein et al., 1995*). This has generated interest in other functional and psychophysical tests that might play a role in the detection of disease, in monitoring or screening (*Hogg & Chakravarthy, 2006*; *Neelam et al., 2009*).

The observation that patients with advanced macular disease, in particular neovascular AMD, often reported their vision as distorted led to the development of the Amsler grid (*Amsler, 1953*) which remains widely used as a means of detecting metamorphopsia (*Faes et al., 2014*; *Keane et al., 2015*). More recently, alternative methods of measuring visual distortions based on various hyperacuities have been developed (*Pitrelli Vazquez & Knox, 2015*). One of these exploits the ability to detect deformations in radial frequency patterns which is highly developed in humans (*Wilkinson, Wilson & Habak, 1998*). While this shape discrimination hyperacuity reaches maturity later than resolution acuity (thresholds decrease, i.e., performance improves, over the first twenty years of life), once adult thresholds are reached they remain stable through adulthood in the absence of pathology, and are relatively resistant to normal ageing (*Wang et al., 2009*). Much of the published data on radial shape deformation thresholds were generated in experiments in which stimuli were presented on computer desktop monitors or in booklet form (*Wang et al., 2002*; *Wang et al., 2009*). More recently the test has been deployed on small mobile devices (*Wang et al., 2013*), a version we will refer to as the handheld radial shape discrimination (hRSD) test.

In this portable form, the test has a number of interesting advantages. Because the platforms (e.g., the Apple iPod touch) on which it is implemented are relatively inexpensive, cost would not be a barrier to making it widely available if the test were shown to be of value. Furthermore, the connectivity of mobile devices, via either wi-fi or mobile phone networks, means that, if patients found the test easy to use, data could be collected by patients themselves in their own homes and then transmitted for remote evaluation. Indeed this was explicitly part of the motivation for developing the test in this form (*Kaiser et al., 2013*; *Wang et al., 2013*) and it has FDA approval for home monitoring in the United States. However, relatively little published information is currently available on the characteristics of the hRSD test.

We therefore wished to investigate the hRSD test, to obtain data on healthy participants and re-examine the effect of age on performance on the iPod version, seeking to confirm the results of *Wang et al. (2009)* for adults. In addition, we investigated test-retest repeatability and participants' views of usability of the test. A three alternative forced choice (3AFC) version of the task, used in a number of current studies (*Lott et al., 2015*; *Pitrelli Vazquez & Knox, 2014*), has recently been superseded by a 4AFC version; so we undertook a direct comparison of 3AFC and 4AFC versions of the test.

## MATERIALS & METHODS

### Ethics statement

This study was performed in accordance with the ethical standards laid down in the Declaration of Helsinki. Experiments of healthy participants were approved by the University of Liverpool Committee on Research Ethics (RETH000827). Anonymised data were also available from small group of older participants from the "Early Detection in Macular Disease" study, approved by the Health Research Authority/NRES Northwest Research Ethics Committee (13/NWEST/0449). Written study information was provided to all participants and written, informed consent obtained.

### Participants

A total of 186 participants were involved in the study, the majority of whom ($N = 170$) were recruited as self-reported healthy participants from staff and students of the University of Liverpool and the wider community. These participants were questioned with regard to their ocular health, and 106 completed slit lamp biomicroscopy and undilated fundoscopy. Participants with either self-reported ocular abnormalities or who were found to have abnormal ocular examinations were excluded from the study. Data were also available from a group of 16 older participants who had neovascular AMD in one eye and no clinical evidence of retinal disease in their fellow, study eye. This was confirmed by spectral domain OCT (Heidelberg Spectralis).

### Procedures

Handheld radial shape discrimination (hRSD) tests were performed using an Apple iPod Touch and the myVisionTrack®(mVT®) application (Vital Art & Science LLC, Richardson, Texas; identical to that used by *Wang et al., 2013*; Fig. 1). Testing was performed with the participant's habitual optical correction. Each eye was tested in turn, with the other eye patched. No specific instruction was given to participants about how they should hold the device; most held it in one hand, in a horizontal orientation, and touched the screen with the index finger of their other hand. In the 3AFC version of the test, two circular and one distorted radial frequency patterns were presented (Fig. 1A). The positions of the distorted and non-distorted patterns were randomised. In the 4AFC version of the test (Fig. 1B), one distorted and three non-distorted patterns were shown. Both the 3AFC and 4AFC versions of the hRSD test followed a 2-down, 1-up adaptive staircase procedure to determine the participant's threshold for detecting distortion.

A subgroup of participants undertook multiple 3AFC hRSD tests to provide test-retest data. 74 were tested twice within a single session, 30 of these participants returned for
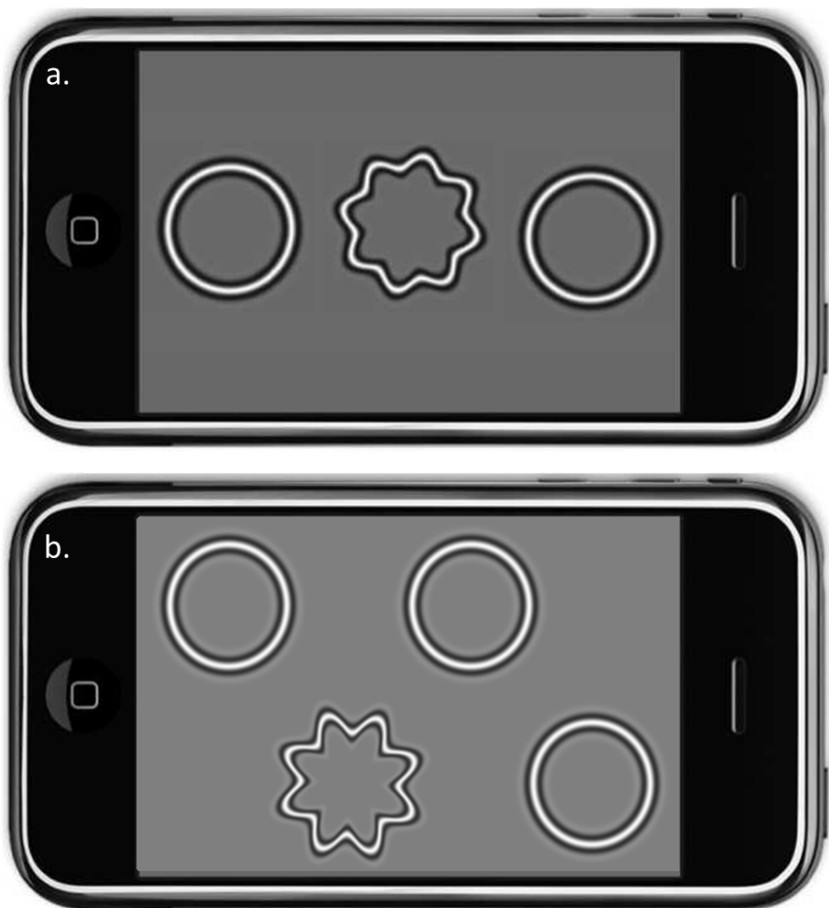

**Figure 1** **The hRSD test.** Simulations of the 3AFC (A) and 4AFC (B) handeld radial shape discrimination stimuli presented on an Apple iPod Touch. In both versions, the position of the target (the distorted pattern) was randomised.

repeat testing within a few months, and 15 after a period of several years. A group of 106 participants were tested with both 3AFC and 4AFC versions of the hRSD test in a single test session. The test order (3AFC vs 4AFC) was counterbalanced across participants and these participants also completed a 5 question usability survey.

Monocular VA ($N = 168$) and contrast sensitivity (CS; $N = 170$) were also measured. Participants' near and distance VA's were measured using 40 cm and 3m ETDRS Logarithmic vision charts. CS was tested using the Pelli Robson Contrast Sensitivity chart at 1m and the results recorded as logCS units.

## Analysis

For the majority of participants, results were available from both eyes. However, as is commonly the case in healthy participants, the data between the eyes were highly correlated. Therefore, where data from two eyes was available, only data from one randomly selected eye were analysed (*Bunce et al., 2014*). Data were collated using MS Excel, and statistical analysis conducted using SPSS.

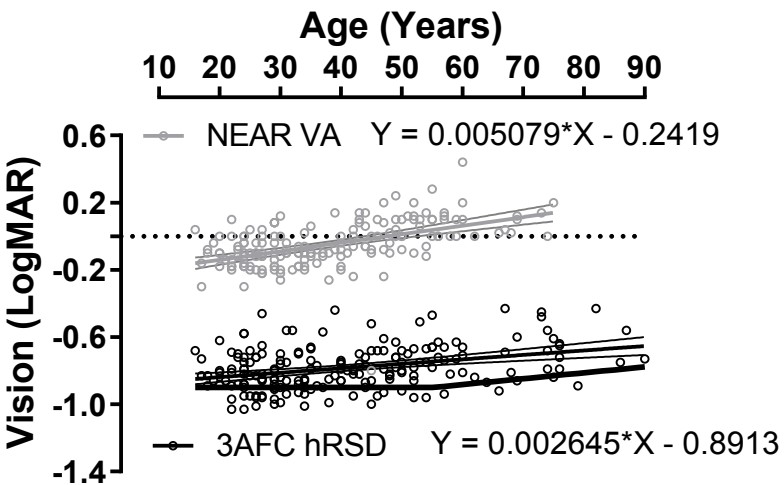

**Figure 2** **Effect of age on hRSD score and near VA.** Plot of 3AFC hRSD thresholds (black circles) and near VA (grey circels; both logMAR) against age. For near VA, data were available for 168/186 participants. Least squares linear regression lines (±95% CI), with equations, are shown. Solid black function at the bottom of the plot is taken from *Wang et al. (2009)*.

## RESULTS

### Characteristics of performance on 3AFC hRSD test

For the 3AFC version of the hRSD test, data were available for 186 healthy, adult participants (72 males, 114 females) who had a mean ± SD age (range) of 42 ± 17 years (16–90 years). Mean (±SD) hRSD threshold was −0.77 ± 0.14 logMAR. Mean near visual acuity (VA; available for 168 of these participants) was −0.05 ± 0.14 logMAR, and contrast sensitivity (CS; $N = 170$) was 1.72 ± 0.12 logCS units.

An analysis of the effect of age on hRSD threshold and near VA for the same eyes (Fig. 2) generated a statistically significant correlation between age and hRSD threshold (Pearson $r = 0.35$; $p < 0.001$) with a slight increase in threshold (worse performance) with age. The slope of the least squares regression line was +0.0026. While this was similar to what was observed for near VA ($r = 0.51$; $p < 0.001$; regression slope: +0.0051), the difference between the two regression slopes was statistically significant ($F_{2,374} = 5.4$; $p = 0.005$). While hRSD threshold across the sample correlated significantly with near VA ($r = 0.21, p = 0.005$), we observed no significant correlation with CS ($r = 0.002, p = 0.8$, Fig. 3).

The intrasession test-retest repeatability of the 3AFC version was investigated in a subgroup of 74 participants (mean ± SD age 43 ± 16y; range 16–80y). The mean (± SD) thresholds for the first and second tests were −0.70 ± 0.22 logMAR and −0.72 ± 0.23 logMAR respectively; there was no statistically significant difference between these thresholds (two-tailed, paired $t$-test; $t = 1.72$; $p = 0.09$). The mean difference between the two tests was 0.02 ± 0.12 logMAR. Data were also available for 30 participants (mean ± SD age 57 ± 24y, range 18–90y) who were tested under identical conditions on two separate occasions, 64 ± 42 days apart. Mean thresholds were −0.68 ± 0.20 logMAR and −0.72 ± 0.18logMAR for the first and second tests respectively ($t = 0.99, p = 0.33$) and the mean

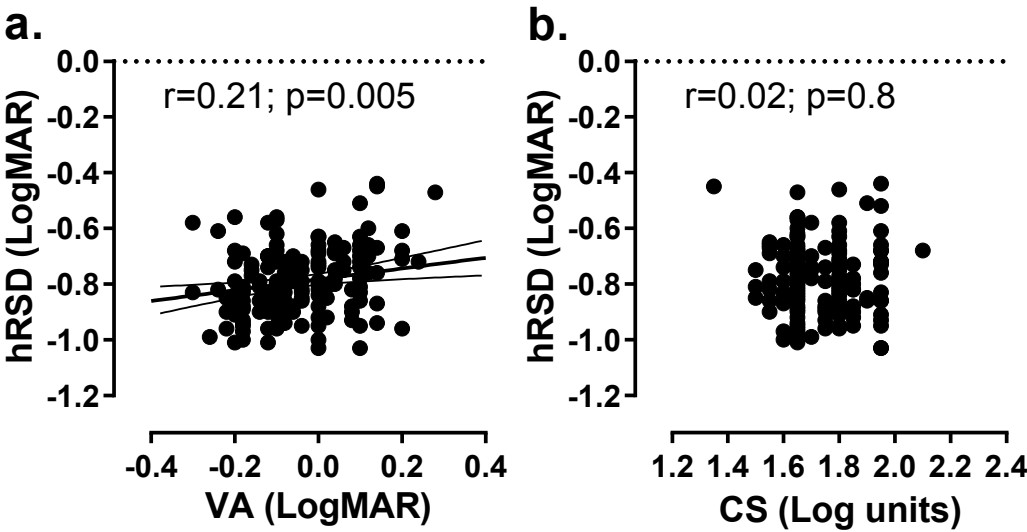

**Figure 3** **Relationship between hRSD threshold and VA and CS.** (A) Relationship between hRSD threshold and near VA. Solid line is the least-squares linear regression line ($\pm$95% CI). (B) hRSD threshold and CS. The correlation coefficient and its statistical significance is also shown on each plot.

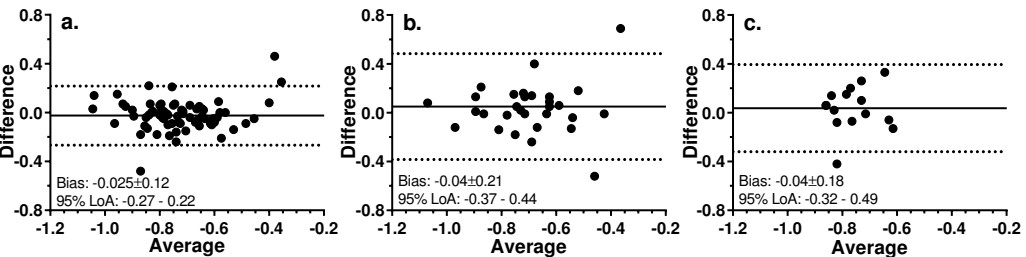

**Figure 4** **Bland-Altman plots for test-retest analysis of 3AFC version of the hRSD test.** Solid line is the mean bias (mean $\pm$ SD also shown); dotted lines are 95% limits of agreement. (A) Two tests run within a single session; $N = 74$. (B) Two tests run 64 $\pm$ 42 days apart; $N = 30$. (C) Two tests run 39 $\pm$ 0.9 months apart; $N = 15$.

difference between the two tests was 0.04 $\pm$ 0.2 logMAR. Finally, 15 participants (mean age 46 $\pm$ 16y, range 18–69y) who were tested on two occasions separated by a mean of 39 $\pm$ 0.9 months. Thresholds for tests 1 and 2 were $-0.74 \pm 0.12$ logMAR and $-0.78 \pm 0.11$ logMAR respectively ($t = 0.78$, $p = 0.44$) and the mean difference between the two tests was 0.04 $\pm$ 0.18 logMAR.

Bland-Altman analysis of all three datasets (Fig. 4), demonstrated the expected low mean bias (numerically equivalent to the mean differences reported above). The 95% limits of agreement were wider when tests were conducted in different sessions separated by either several weeks (Fig. 4B) or years (Fig. 4C) compared to within the same session (Fig. 4A). There was no obvious pattern to the scatter of the points on the plots.

## Comparison of 3AFC and 4AFC results

The 3AFC and 4AFC versions of the hRSD test were compared in 106 participants (42 males, 64 females; mean age 36 $\pm$ 13 years, range 18–69 years). All of the participants
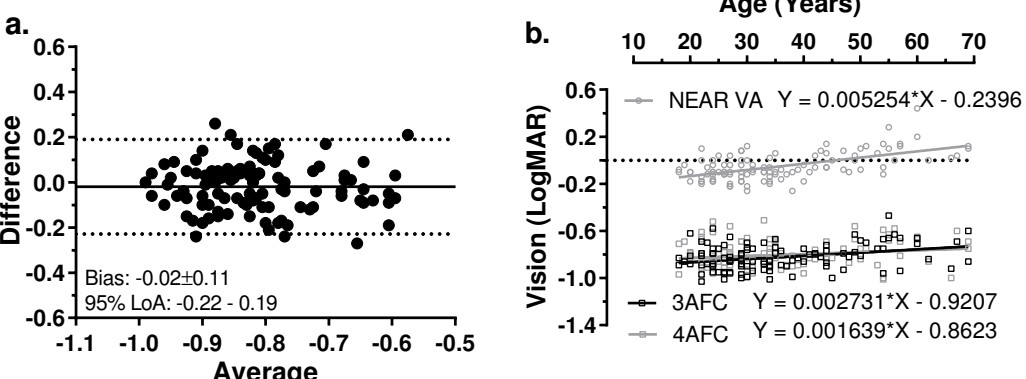

**Figure 5   Comparison of 3AFC and 4AFC versions of the hRSD test.** (A) Bland-Altman plot showing Bias and 95% LoA for 3AFC vs 4AFC version of the the hRSD test. (B) Influence of age on 3AFC and 4AFC results, and on near VA for 106 participants. Least squares linear regression lines (with 95% CI) along with the functions for the lines are shown, as well as the regression parameters.

had a normal, undilated fundoscopy examination in the selected study eye. Mean 3AFC and 4AFC hRSD thresholds were $-0.82 \pm 0.11$ logMAR and $-0.80 \pm 0.12$ logMAR respectively ($t = 1.85, p = 0.067$), mean difference was $-0.02 \pm 0.11$ logMAR. Bland-Altman analysis (Fig. 5A) revealed a very low level of bias ($-0.02 \pm 0.11$ logMAR) and 95% limits of agreement of $-0.22$ to $0.19$ logMAR. We compared the effect of age on both the 3AFC and 4AFC hRSD thresholds and near VA (Fig. 5B) and observed a weak, though statistically significant correlation between age and 3AFC thresholds ($r = 0.32, p = 0.0009$) and between age and near VA ($r = 0.33, p < 0.0001$), but not between age and 4AFC thresholds ($r = 0.18. p = 0.061$). While the regression slopes for the 3AFC and 4AFC thresholds were very similar (and not statistically significantly different, $F_{1,208} = 0.86, p = 0.35$), the slopes for the regression line of the combined hRSD data and that of near VA were significantly different ($F_{1,314} = 9.76, p = 0.002$). The mean test times for the 3AFC and 4AFC versions were $193.3 \pm 56.5$s and $194.2 \pm 54.9$s respectively (paired $t$ test; $t = -0.158, p = 0.875$).

The results of the usability questionnaire demonstrated that the majority of participants understood how to use the test, found it easy to use, felt it did not take too long and were confident using it (Table 1). We also asked participants their views comparing the 3AFC and 4AFC versions of the test. While 41.5% expressed no preference between versions, 33.9% found the 3AFC version easier to use, leaving 24.6% who found the 4AFC version easier to use.

## DISCUSSION

The 3AFC hRSD threshold that we observed in large sample of healthy adult participants ($-0.77 \pm 0.14$ logMAR), is slightly poorer than that reported by *Wang et al. (2009)*. They investigated both children and adults, with the stimuli presented in two different formats, a 2AFC version presented on a computer monitor, and a 4AFC version presented on cards. The thresholds returned from these two methods were in good agreement. For adults aged 21–55y ($N = 59$) they reported a threshold of $-0.91 \pm 0.1$ logMAR. However, the same

**Table 1  Usability questionnaire results.** Participant responses to each statement, shown as % of participants.

|  | Strongly disagree | Disagree | Neutral | Agree | Strongly agree |
|---|---|---|---|---|---|
| I understood how to use the hRSD device | 2 | 0 | 1 | 18 | 79 |
| The hRSD test was easy to use | 1 | 1 | 5 | 15 | 78 |
| The hRSD test did not take too long to do | 1 | 3 | 4 | 37 | 55 |
| I could use the hRSD device to test my own vision | 1 | 1 | 11 | 30 | 57 |

group subsequently reported slightly poorer performance with the handheld, iPod version of the task ($-0.69$ logMAR) for a control group of older healthy participants ($N = 27$, mean age 69y) in a clinical study (*Wang et al., 2013*). This corresponds well to a median of $-0.71$ logMAR recently reported for the same version of the task, obtained from a group of healthy participants with a mean age of 73y (*Lott et al., 2016*). Given differences in participant selection, testing procedures and study settings, the values for hRSD thresholds appear to be consistent across these studies.

The slight increase in hRSD threshold with age which we observed is also consistent with that reported previously (Fig. 2; *Wang et al., 2009*). While the correlation between age and hRSD threshold was statistically significant, the slope of the regression line implied an average decline of only 0.026 logMAR per decade between the ages of 20y and 80y. Given that the decline in near VA over the same period proceeded at almost twice this rate, hRSD performance appears to be relatively resistant to the effects of normal ageing. It has been suggested that radial shape discrimination might be stable up to the fifth decade, with a slight decline then occurring, even in the absence of any obvious pathology (the function represented by the solid black line in Fig. 2 taken from *Wang et al. (2009)*. While we saw no evidence of this pattern in our data it should be noted that our oldest participants were selected on the basis that they had no retinal pathology in their selected eye. As early AMD is a highly prevalent condition, our participants are probably unrepresentative of the general population at older ages into the seventh and eighth decades of life. More data on the performance of older participants with early and intermediate AMD would be useful. In patients with intermediate AMD (large retinal drusen and/or pigmentary changes) it has been reported that the average threshold of the 3AFC iPod test drops to $-0.37$logMAR (*Wang et al., 2012*).

The mean difference observed between two hRSD tests within a single session, and separated by several weeks or even many months was consistently close to zero and the limits of agreement relatively narrow. Even over an extended period of time, performance remained stable, although we were only able to assess this in a relatively small (and
relatively young) group of participants. The Bland-Altman plots did not suggest any consistent relationship between average performance and variability. Low levels of test-retest variability are a desirable feature of a test that might be used longitudinally over extended periods to monitor for the development of disease. However, again given that the older eyes in this study were free from even early signs of AMD, it remains to be established what level of change from baseline in older eyes might be expected in eyes with early/intermediate AMD and how this compares with inherent test-retest variability (*Wang et al., 2013*).

There is very little published data on the 4AFC version of the hRSD test. There is one report on data from 10 relatively young control participants (mean age 35.5y, range 16–66) from a clinical study, in which the task was presented on an iPad reporting a threshold of $-0.70 \pm 0.10$ logMAR compared to our $-0.80 \pm 0.12$ logMAR (*Bennett et al., 2016*). We found no differences in age effects between the 3AFC and 4AFC versions of the test, or in the testing time. This absence of a detectable difference is useful for two reasons. While statistically we would have expected the thresholds measured by the two versions of the task to be similar, psychological or strategic factors might have played a role in participant performance (*Jäkel & Wichmann, 2006*). For example, anecdotally, we have found that in parallel clinical studies some older participants occasionally become confused about the positioning of the stimuli on the screen when using the 3AFC version. The three patterns are presented in a line, with their vertical positions (when the test is oriented horizontally) randomized (Fig. 1A). Occasionally, older participants have picked a pattern based on its apparent height on the screen rather than because it is distorted. While we have detected issues like this whenever they have occurred, this sort of behaviour might be a problem in a busy clinic as opposed to research studies. However this issue does not apply to the 4AFC version of the test (Fig. 1B). In addition, confirming that the two versions provide comparable thresholds is useful as it suggests that it is possible to make direct comparisons between older studies based on the 3AFC version and newer studies based on the 4AFC version.

We found no difference in the usability of the two versions. Similar to previous results (*Wang et al., 2013*), all of our participants found the test easy to understand and execute. No participant failed to produce a result. However, since this is a study of motivated healthy volunteers, further data on test usability needs to be reassessed with participants drawn from groups in which this test might have a clinical role (e.g., patients at risk of AMD and other macular pathologies).

## CONCLUSIONS

The hRSD test has shown clear promise for the detection and monitoring of a number of retinal pathologies (*Bennett et al., 2016*; *Wang et al., 2013*). We have confirmed in healthy participants that it produces consistent results both over time, and indeed across studies in different settings. Presented on small mobile devices (e.g., the iPod and iPad, smartphones), it is easy to use and data can be easily stored and transmitted for analysis. This study provides the largest sample of normative data for the 3AFC and 4AFC hRSD thresholds in the current literature, and serves as a baseline for future studies.

## ACKNOWLEDGEMENTS

We are grateful to all of the participants who took part in this study, and Vital Art & Science LLC for software support.

### Funding

Part of this work was funded by a grant from the Dunhill Medical Trust (R283/0213). The funders had no role in study design, data collection and analysis, decision to publish, or preparation of the manuscript.

### Grant Disclosures

The following grant information was disclosed by the authors:
Dunhill Medical Trust: R283/0213.

### Competing Interests

The authors declare there are no competing interests.

### Author Contributions

- Jae Y. Ku conceived and designed the experiments, performed the experiments, analyzed the data, wrote the paper, prepared figures and/or tables, reviewed drafts of the paper.
- Ashli F. Milling conceived and designed the experiments, performed the experiments, analyzed the data, reviewed drafts of the paper.
- Noelia Pitrelli Vazquez performed the experiments, analyzed the data, reviewed drafts of the paper.
- Paul C. Knox conceived and designed the experiments, analyzed the data, contributed reagents/materials/analysis tools, wrote the paper, prepared figures and/or tables.

### Human Ethics

The following information was supplied relating to ethical approvals (i.e., approving body and any reference numbers):

University of Liverpool Committee on Research Ethics - RETH000827.

Health Research Authority/NRES Northwest Research Ethics Committee - 13/NWEST/0449.

### Data Availability

Figshare: https://figshare.com/s/4bf6784d1ece3b8b2938.

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
