# Peer review of "Performance, usability and comparison of two versions of a new macular vision test: the handheld Radial Shape Discrimination test"

_PeerJ, doi:10.7717/peerj.2650_

## Round 0.1 · original submission · Minor Revisions

Two reviewers have read your MS and find it acceptable. They offer suggestions for minor revisions, which I hope you will enact and otherwise the manuscript can be Accepted.

Reviewer 1 ·

Basic reporting

The article is well-written, clear, and detailed.
Minor comments:
I’m not following the thought process in lines 54-56 as written: Because the hRSD is easy to use and inexpensive, clinical data is needed?
Line 217 “The usability questionnaire demonstrated…” should be “The results of the usability questionnaire demonstrated…”
Line 231 There is an extra comma in “handheld, iPod version of the”
Table 1 shows percentages, right? Not labelled. Some rows don’t add up to 100 (the first row is < 94), so this is not obvious. Some participants appear to have agreed that the device was easy to use, but did not answer whether or not they understood how to use the device.

Experimental design

No major comments.
Minor comment: One methodological factor that might add noise to the results—that participants were not instructed in precisely how to hold the devices—probably mimics how this test would be used in practice. I would be interested to know if participants held the device the same way on retest, and whether failing to do so would dramatically alter their performance. If so, it would be desirable to standardize the instructions for how to hold the iPods are held during testing.

Validity of the findings

No major comments.
Minor comments:
The authors state on line 290 that they found no difference in the usability of the two versions, but point to an (admittedly anecdotal) difference on line 278, which they suggest did not play out in their data directly because of the way the experiment was conducted. Specifically, some participants were confused about the positioning of the 3AFC stimuli and the authors clarified the instructions in a way they believe would not be done in a clinical setting. This points to a usability difference.

Additional comments

This was a pleasure to read.

·

Basic reporting

Article has met these basic reporting standards. Figures are relevant to the content of the article. The cited literature is appropriate and supports the body of the research. The article includes sufficient introduction and background.

Experimental design

This study is nicely done to validate data in the largest sample for the 3AFC and 4AFC hRSD thresholds. To this levels, the data and hence conclusion is solid. However, the validity of the approach needs to be tested in a larger population of patients with retinopathy and more specifically macular disease. At this point it is not clear if hRSD is as sensitive or specific for early detection of macular disease (Compared to other diagnostic tests such as Amsler grid, etc... ).

Validity of the findings

The study is statistically acceptable. And the conclusion is well supplemented with graphs, figures and tables.

Additional comments

Now that the authors established the baseline thresholds for 3AFC and 4AFC hRSD for future studies, I strongly encourage the authors to design a new study and validate the usability and also efficacy of hRSD in patients with visually disabling macular disease.

---

## Round 0.2 · accepted · Accept

Thank you for making the necessary revisions to this study. I am happy to accept the manuscript.